# Effect of Occupational Stress on Pharmacists’ Job Satisfaction in Saudi Arabia

**DOI:** 10.3390/healthcare10081441

**Published:** 2022-07-31

**Authors:** Lamees Aldaiji, Ahmed Al-jedai, Abdulrahman Alamri, Ahmed M. Alshehri, Nouf Alqazlan, Yasser Almogbel

**Affiliations:** 1Department of Pharmacy Practice, College of Pharmacy, Qassim University, Buraidah 51452, Saudi Arabia; lameesabdullah12@gmail.com (L.A.); noufvvvv@gmail.com (N.A.); 2Deputyship of Therapeutic Affairs, Ministry of Health, Riyadh 12553, Saudi Arabia; ahaljedai@moh.gov.sa; 3Colleges of Medicine and Pharmacy, Alfaisal University, Riyadh 11211, Saudi Arabia; 4Pharmaceutical Care Services, Ministry of the National Guard-Health Affairs, Riyadh 11426, Saudi Arabia; amriab@ngha.med.sa; 5King Abdullah International Medical Research Center, Riyadh 11426, Saudi Arabia; 6College of Pharmacy, King Saud bin Abdulaziz University for Health Sciences, Riyadh 11426, Saudi Arabia; 7Department of Clinical Pharmacy, College of Pharmacy, Prince Sattam Bin Abdulaziz University, Al Kharj 16273, Saudi Arabia; ah.alshehri@psau.edu.sa

**Keywords:** pharmacist, work stress, satisfaction, pharmacy profession

## Abstract

Work stress occurs when employees have to deal with pressures that do not align with their skills, knowledge, or expectations. This study aimed to assess the impact of work stress on job satisfaction among pharmacists in Saudi Arabia. Therefore, a cross-sectional, self-administered, paper-based survey was conducted between August 2019 and October 2020 using three scales. Descriptive and analytical statistical analyses were performed. A linear regression analysis was used to determine the relationship between occupational stress and job satisfaction among Saudi pharmacists. A total of 284 questionnaires were completed. Multiple linear regression analyses showed a significant negative relationship between occupational stress and job satisfaction (β = −0.456, 95% CI, −0.561 to −0.350), a positive relationship between confirmation and satisfaction (β = 0.147, 95% CI, 0.005 to –0.290), and a negative relationship between working in hospitals and job satisfaction (β= −3.009; 95% CI, −5.424 to −0.593) when other variables were kept constant. The results of this study indicated that occupational stress and satisfaction negatively influenced pharmacists, whereas confirmation was associated with better satisfaction. Moreover, hospital pharmacists had lower job satisfaction. The job satisfaction of pharmacists may help improve medication safety and ensure an adequate pharmacist workforce.

## 1. Introduction

Pharmacists are essential elements of the healthcare industry [1]. Pharmacists can perform many tasks, including dispensing prescription medications, monitoring medication therapies to avoid adverse drug reactions and interactions, providing drug information to healthcare professionals, and ensuring the safe use of medications [1]. In the last century, a pharmacist’s job description was to dispense and prepare medications. Nowadays, the scope has widened to include many specialties and to ensure efficient participation in the patient care process, which has helped reshape the pharmacist’s job perception and recognition.

Pharmacists in Saudi Arabia practice their role in different settings. The community and hospital pharmacists perform their duty of reviewing and dispensing prescribed medications. Furthermore, pharmacists counsel patients. Clinical pharmacists provide clinical services and participate with the healthcare team [2]. Moreover, the pharmacist has a significant rolein regulatory agencies, including Good Manufacturing Practice (GMP) and Saudi Food and Drug Authority (SFDA), which took the lead in regulating and marketing pharmaceutical products in Saudi Arabia. In addition, pharmacists have a significant role in pharmaceutical companies as medical representatives, scientific officers, industrial technologists, and compounding manufactured products. Academic institutions are the root of preparing pharmacists for the market. Colleges of pharmacy provide different educational and promotional degrees to suit the market needs.

Pharmacists endure significant professional stress, which could lead to job burnout [3,4]. Work stress is defined as a state in which one or more variables interfere with a worker’s physical, psychological, or social stability [5]. Stress can be divided into two types: positive and negative. Positive stress is referred to as eustress, and negative stress is referred to as distress. Work performance may be stimulated and improved by eustress, and it can motivate employees to work harder. Distress can have negative consequences, affecting workers’ health and productivity and directly impacting the organisation’s performance [6]. Worry and exhaustion are common symptoms of stress [7]. Workers under a high level of stress at work may become discouraged, less productive, and less safe in the workplace [8]. Furthermore, healthcare professionals may experience occupational stress due to inequalities in job demands, patient accountability, professional stigma, skills, income, workplace climate, available resources, and other organisational concerns such as appreciation [9,10]. Although being a pharmacist is considered rewarding, it can be accompanied by stressful conditions (long hours, heavy workloads), which, apart from harming the pharmacists’ health, makes it more difficult for them to serve to the best of their abilities. In addition, stress can harm the quality of services provided, resulting in a loss of the trust of business beneficiaries and in bad financial performance [11,12]. Work stress is responsible for 60–80% of all workplace accidents [13]. Many studies and reports demonstrate that many countries, including Saudi Arabia, consider working in the pharmaceutical sector as stressful and demanding, requiring a high degree of skill and nearly limitless patience [14,15].

Job satisfaction can be defined as the level of favourableness that encourages and motivates employees to work better and be more productive [16]. However, it is directly related to an individual’s behaviour in the workplace [17]. In addition, satisfaction affects staff retention and commitment, further influencing a pharmacist’s decision to quit their profession [18]. In 2001, Motoko and Michiko identified five elements of workplace satisfaction: (1) job interest; (2) expectations; (3) workload; (4) health and welfare benefits; and (5) career progression [19]. Barnett and Kimberlin reported that job satisfaction in the pharmacy field should be considered for three reasons: (1) job satisfaction benefits the employer since satisfied employees are less likely to be absent, participate in behaviour that is detrimental to the organisation, or quit jobs; (2) job satisfaction has a positive impact on patients since it is linked to greater performance among pharmacists; and (3) job satisfaction benefits pharmacists since it improves their physical and mental health [20].

The number of pharmacy colleges in Saudi Arabia has dramatically increased during the last 16 years, from one college established before more than fifty years to more than twenty colleges. That was a reason for the increasing competition in finding jobs for pharmacists. Thus, in Saudi Arabia, an effort has been placed with vision 2030, to increase the employment rate of Saudi Arabian pharmacists and improve healthcare services provided to society [21,22,23,24]. The Saudi government supports the privatization of the health sector to improve the quality of the provided services. It is essential to know how satisfied pharmacists are in Saudi Arabia and whether they suffer from job stress or overload. Retention of employees will stabilize the working environment and decrease medical errors. Furthermore, retention will reduce the new employees’ training bill. In 2004, a study conducted among 1737 American pharmacists found that over 65% of pharmacists were satisfied with their jobs. However, more than 70% suffered from job stress and role overload [12]. Many studies have focused on job stress among pharmacists [14,25,26]. However, no study has investigated the relationship between job stress and satisfaction among Saudi pharmacists. Therefore, this study aimed to assess the impact of stress on pharmacists’ job satisfaction in Saudi Arabi.

## 2. Materials and Methods

### 2.1. Study Design and Data Sources

This cross-sectional study was conducted to determine the effect of occupational stress on job satisfaction among pharmacists. The surveys were paper-based and self-administered. The data were a part of a large survey collected from Saudi pharmacists in different sectors across Saudi Arabia, including academic institutions, hospitals, the drug industry, and conferences, between August 2019 and October 2020. A non-probability convenient sampling technique was used to identify the sample. All licensed pharmacists working in Saudi Arabia were eligible to participate. A trained research approached pharmacists at each location and asked them to participate after describing the study goals. Informed consent was obtained from each participant before commencing data collection. The participants were informed to drop the completed survey in an envelope. The G*Power software (version 3.1.9) was used to compute the sample size. The sample size was computed to be 257 participants. The Regional Research Ethical Committee (Institutional Review Board) granted permission to conduct this study.

### 2.2. Informed Consent

An oral explanation of the objectives of this study was provided to the participants by one of the authors. The data collector asked pharmacists to complete the survey along with the consent form, which was completed before participation. All participants were assured of strict confidentiality and were free to withdraw from the study at any time.

### 2.3. Survey Design

The survey included four sections: sociodemographic, occupational stress, an expectation-confirmation assessment, and job satisfaction. The survey was initially drafted in English and then translated and conducted in Arabic. A back-translation process was performed by independent bilingual translators, who translated the questionnaire from English to Arabic [27]. A questionnaire translated into Arabic from a previously published study (21 items) was utilised, which matched the short form of the Effort-Reward Imbalance (ERI) scale (16 items) [28]. An Arabic version of the ERI questionnaire was translated and validated for this study. The translated version was reviewed by two pharmacy practice specialists and validated for ten participants. The questionnaire and short version can be found in the Appendix A.

### 2.4. Variables

#### 2.4.1. Independent Variables

Independent variables included sociodemographic variables, the pharmacists’ occupational stress variables, and the pharmacists’ expectation-confirmation assessments. Sociodemographic variables included various demographic and social variables. Demographic variables included age, gender, marital status, number of children, and chronic disease. Social variables included total monthly income, working hours per week, years of pharmacy work experience, level of pharmacy education (pharmacy bachelor’s degree, doctor of pharmacy, pharmacy master’s degree, pharmacy residency, and doctor of philosophy in pharmacy), and occupational variables (hospital pharmacist, clinical pharmacist, community pharmacist, medical representative in a pharmaceutical company, industrial pharmacist, and university professor).

Occupational stress among the pharmacists was measured using the Arabic version of the ERI scale. The scale included 16 items divided into three scales: extrinsic effort, extrinsic reward, and over-commitment. Psychosocial workload was used to evaluate the extrinsic effort, which identified the pharmacists’ financial situation (e.g., wages), self-esteem, and work opportunities (e.g., promotion prospects and job security). The ERI scale was measured by six items (items ERI1-ERI6) on a 4-point Likert scale (from 1 = strongly disagree to 4 = strongly agree), with higher ratings indicating the need to increase efforts. The extrinsic reward was measured using ten items (items ERI7-ERI16) on a 4-point Likert scale [28]. Over-commitment is described as a combination of attitudes, behaviours, and feelings that show excessive striving as well as a strong need for acceptance and esteem as a personal (intrinsic) component [28].

The pharmacists’ expectation confirmation was measured using ten items on a 7-point Likert scale item (from 1 = strongly disagree to 7 = strongly agree), which is based on the Expectation-Confirmation Theory (ECT); a cognitive theory that seeks to explain post-purchase or post-adoption satisfaction as a function of expectations, perceived performance, and disconfirmation of beliefs. The ECT has been extensively used as one of the primary theories in the marketing literature, which was initially established by Oliver (1980) and further developed by Bhattacherjee (2001) in response to a study on the cognitive behaviour of continuation intention in the consumer marketing sector [29]. A pharmacist’s confirmation is the combination of expectation and experience. A confirmation is positive when expectation is equal to or less than the experience [29]. However, a confirmation is negative when the expectation is exceeds the experience. The confirmation was measured by subtracting the total score of perceived performance (five questions) from the total score of expectations (five questions).

#### 2.4.2. Outcome Variables

The study outcome variable was the pharmacists’ job satisfaction, which was adapted from Bhattacherjee and Premkumar’s survey (2004) and measured using four items on a 7-point Likert scale [30]. The pharmacists’ job satisfaction was defined as when the pharmacists liked (satisfied) their job or disliked (dissatisfied) their job.

### 2.5. Statistical Analyses

Descriptive analyses were conducted to investigate categorical and continuous sociodemographic variables and stress levels. Two hundred fifty-seven participants was calculated as the sample size, using G*Power, assuming that the number of predictors was fifteen, at 0.05 alpha level, 0.99 power, and 0.15 middle effect size. All independent variables with a *p*-value < 0.2 were added to the multiple linear regression analysis to control for any statistically insignificant confounding variables [30,31]. All data were encoded and entered into Microsoft Excel 2016, and statistical analyses were performed using STATA 16.

## 3. Results

A total of 371 questionnaires were distributed, and 284 were returned. Descriptive analyses were conducted to provide an overview of the study sample (Table 1). The mean age of the participants was 33.4 ± 6.5 years, and their average work experience was 8 ± 7.1 years. The average monthly income for the sample was USD 4364.5 ± USD 1829, and the total working hours per week were 41.2 ± 10.2 h.

Overall, most respondents were male (61.1%) and married (62.8%), whereas less than half (46.8%) reported having no children. Among the respondents, 17.6% reported having been diagnosed with a chronic disease. The majority of the participants (78.5%) had a bachelor’s degree in pharmacy and worked in hospitals (80.6%).

Table 2 shows the descriptive statistics of the scales that have been used in this study. Cronbach’s alpha was more than 0.7, which is considered acceptable reliability [32].

Table 3 describes the association between the pharmacists’ job satisfaction and other predictors using a univariate linear regression analysis. The pharmacists’ job satisfaction was shown to have a significant negative relationship with the pharmacists’ occupational stress (β = −0.407; 95% CI = −0.504 to −0.310; *p* < 0.001) and a positive relationship with the pharmacists’ expectation confirmation (β = 0.318; 95% CI = 0.207 to 0.4288; *p* < 0.001). Regarding age and gender, we also found a significant positive relationship between the pharmacists’ job satisfaction and age (β = 0.164; 95% CI = 0.051 to –0.276; *p* = 0.004) and the male gender (β = 2.580; 95% CI = 1.163 to –3.998; *p* < 0.001). In addition, there was a positive relationship between the pharmacists’ job satisfaction and being married (β = 1.48; 95% CI = 0.009 to –2.950; *p* = 0.049) and having a higher level of education (β= 1.788; 95% CI = 0.057–to 3.518; *p* = 0.043). Finally, there was a negative relationship between the pharmacists’ job satisfaction and working in a hospital (β = −2.176; 95% CI = −4.182–to −0.169; *p* = 0.034).

A multivariate linear regression analysis was performed to account for confounders (Table 4). A significant negative relationship was found between the pharmacists’ job satisfaction and occupational stress (β = −0.456; 95% CI = −0.561 to −0.350; *p* < 0.001) and with the pharmacists working in a hospital (β = −3.009; 95% CI = −5.424 to −0.593; *p* = 0.015) when other variables were kept constant. In contrast, there was a significant positive relationship between the pharmacists’ job satisfaction and confirmation (β = 0.147; 95% CI = 0.005 to –0.290; *p* = 0.041) when other variables were kept constant.

## 4. Discussion

Pharmacists must ensure the safety of medications supplied to patients seeking medical advice as a part of their efforts to improve pharmacy service quality. Furthermore, they must work within the framework of numerous hospital accreditations, licencing standards, and a national health insurance system’s requirements [4]. Moreover, the pharmacists’ job satisfaction directly influences medication dispensing safety, which significantly affects the quality of patient care [33]. Therefore, employees who are stressed, depressed, or dissatisfied cannot produce the same quality of work and performance as those who are satisfied and less stressed [6], which makes this an urgent problem that must be addressed. Therefore, this study aimed to investigate the effects of occupational stress on job satisfaction among pharmacists in Saudi Arabia. We found that pharmacists’ job satisfaction in Saudi Arabia was negatively associated with working in a hospital and occupational stress and positively associated with confirmation.

The mean age of the participants was 33 years, which means that most participants were young pharmacists (from their early twenties to their mid-thirties). This might be because the number of pharmacists graduating from public schools in Saudi Arabia has increased rapidly from 150 to 250 graduates in 2000 to 945 graduates in 2015 [34]. Based on these data, the average work experience of our participants was approximately eight years. In contrast, the pharmacists’ average working hours per week were 41.24 h/week. According to Saudi labour regulations, ‘When the employer uses the daily system, the worker should not work more than eight hours per day; when the weekly system is used, they should not work more than forty-eight hours a week.’ [35].

Stress is one of the main factors influencing the performance of healthcare workers, including pharmacists [36]. Pharmacists experience stress and exhaustion due to prolonged exposure to demands, limited resources, and decreased cognitive abilities [37]. Burnout is a stress response, especially when people are disturbed or have difficulties [38]. Furthermore, if the workload exceeds the standard, the employee is likely to become overwhelmed, leading to hazards such as burnout and subsequent breakdowns, bad feelings, and dissatisfaction, ultimately causing them to leave the job [39]. A study of hospital and community pharmacists in Northern Ireland found that work interruptions, heavy workloads, and insufficient staffing were the most stressful aspects of a pharmacist’s job [40]. The final multiple linear regression analysis showed a negative association between the pharmacists’ job satisfaction and stress. These outcomes are consistent with the findings of previous studies that have demonstrated a negative correlation between stress and work satisfaction. A Chinese study of healthcare practitioners that investigated the relationship between satisfaction and other factors found that work satisfaction was significantly associated with work stress, work-family conflicts, and sociodemographic factors [41]. A survey conducted among 129 nurses in Australia to measure nurses’ stress and satisfaction found that lower levels of stress led to a higher level of work satisfaction and lower mood disturbances [42]. The vast majority of research on reducing personal stress demonstrates changes in lifestyle, regular exercise, yoga, meditation, interpersonal communication, consulting a mentor, delegating assigned tasks, reducing tasks, and using time-management strategies [43].

The ECT linked confirmation to satisfaction. An expectation precedes any event, and satisfaction occurs if an expectation is met positively. In contrast, dissatisfaction occurs when an expectation is not met positively [44]. Furthermore, in the context of employee work, Porter and Steers (1973) defined met expectations as the difference between what one expects to experience in the job and what one experiences in the job [45]. According to this theory, the degree of disconfirmation, or the degree to which expectations are not satisfied, could impact satisfaction. Disconfirmation theory states that contentment is reduced when experiences fall short of expectations—a disappointment effect. Expectations positively influence satisfaction when experiences exceed expectations; this is known as the positive surprise effect [46]. Accordingly, we found a significant positive relationship between satisfaction and confirmation, which is not unexpected because it is usually affected [29].

Job dissatisfaction among pharmacists has been linked to their work in several studies [47,48]. Pharmacists working in hospital pharmacies report higher levels of job satisfaction than those working in other practice settings [42,43]. Previous studies found varying degrees of work satisfaction among hospital pharmacists. For example, 67% of those in India [49] and 67.2% of those in the USA [21] were satisfied with their jobs. In 2021, a study conducted in Saudi Arabia among 437 pharmacy students in their final year demonstrated that hospital pharmacy was the most preferred career and community pharmacy was the least preferred [22]. In contrast, a recent Ethiopian study involving 232 pharmacists working in hospitals found that job satisfaction among hospital pharmacy professionals in Ethiopia was extremely low [50]. Since the present study included all pharmacists working in various pharmaceutical sectors, it examined the job stress and satisfaction among different sectors. A negative association between the pharmacists’ job satisfaction and hospital satisfaction was found, which means that healthcare professionals were less satisfied than other pharmacy sector workers. This could be due to the higher work expectations from most Saudi hospital pharmacists than those in other pharmacy sectors (e.g., community pharmacy), which included a majority of non-Saudi pharmacists [51]. Similarly, a previous study conducted among hospital pharmacists in Saudi Arabia indicated that less than 50% of responding pharmacists were satisfied with their jobs [52]. In contrast, another study found that in Jordan, pharmacists working in community pharmacies had lower job satisfaction than pharmacists working in hospital pharmacies [48].

Additionally, according to Lee et al. [53], hospital pharmacists in South Korea have lower job satisfaction. In South Korea, hospital pharmacists represent only 11.7% of actively practising pharmacists, whereas community pharmacists represent 73.1% [54]. Most pharmacists in our study worked in hospitals (80.59%), possibly due to the higher availability of jobs in hospitals in Saudi Arabia in previous years. Furthermore, Saudi employees prefer to work in hospitals due to the higher job security. Moreover, most pharmacists (employed in the public sector) have permanent jobs and are less competitive. A descriptive cross-sectional study conducted in India to measure work satisfaction among Indian pharmacists reported that the fear of losing one’s job was a key factor contributing to job dissatisfaction [55]. Nevertheless, regarding the Saudi Arabia 2030 vision, we believe that more fresh pharmacists will choose to be employed in private sector companies to gain more income and have greater chances of acquiring jobs in the private sector than in government hospitals [56].

This study has a few limitations. The parameters of the cross-sectional survey limit the size and scope of the study. Cross-sectional studies may have limited applicability in establishing the causality but not the association. Researchers have examined the specific relationships that occur at a certain point in time. A five pages paper-based survey was utilized in the current investigation. Participants may skip some questions. As a result, missing values may have influenced the survey results, particularly the responses to questions at the end of the questionnaire. Based on the participant’s right, participants can drop out at any time, and they cannot be forced complete the questionnaire. Furthermore, it is not allowed for the participants to review their response after submitting the survey. Despite our assurances to the participants about their confidentiality, they may have been reserved in their answers due to social factors. Some participants may omit specific questions they do not believe relevant or have a legitimate resolution. This study followed a non-probability convenient sampling technique. Due to the rejection rate, this study could not focus on some sectors, such as hospitals. Thus, investigators had no control over the sampling and distribution between participants. These findings are generalizable to only a similar population.

To the best of our knowledge, this study is the first to investigate the effect of occupational stress on job satisfaction among Saudi pharmacists. Job satisfaction and the associations between job satisfaction and work stress confirmation were investigated among pharmacists working in different sectors. Improving pharmacists’ working conditions may decrease their stress levels and encourage their development. Working conditions can be improved by reducing working hours or at least by not exceeding the standard working hours per week. Furthermore, salary system improvement and pharmacist consideration are factors for improving pharmacist job satisfaction. Improving shift working hours, rest breaks, and shift patterns can also enhance job satisfaction. Providing proper training, an overview of any new technology, introduction to the workplace and to changes to administrative culture, and a summary of any company rules or work capacity modifications may increase job satisfaction. Those factors may help decrease stress and improve satisfaction among pharmacists. These findings can encourage future research. Further exploration is warranted to determine the impact of pharmacist satisfaction on performance, patient safety, and organisational performance and stability.

## 5. Conclusions

This study aimed to investigate the effects of occupational stress on job satisfaction among pharmacists in Saudi Arabia. We found that working in hospitals and occupational stress negatively affected job satisfaction. In addition, satisfaction positively impacted confirmation, and an increase in satisfaction lead to better pharmacist confirmation. This study may encourage further research on pharmacists’ stress management and satisfaction in other healthcare sectors, especially in Saudi Arabia, and bring the current practice to a higher level.

## Figures and Tables

**Table 1 healthcare-10-01441-t001:** Pharmacists’ Sociodemographic Characteristics in Saudi Arabia (*n* = 284).

Characteristics	Number of Participant (*n*)	Percentage (%)
Age (years); mean (±SD)	33.4 (±6.5)	
Gender		
Male	170	61.1
Female	108	38.8
Marital status		
Married	176	62.8
Non-married	108	38.0
Having children		
Yes	151	53.1
No	133	46.8
Chronic disease		
Yes	50	17.6
No	234	82.3
Education level		
Bachelor’s level degree	223	78.5
Graduate-level degrees	61	21.4
Pharmacy job setting		
Hospital pharmacist	204	74.1
Academic institutes	20	7.3
Pharmaceutical marketing	18	6.6
Pharmaceutical industry	10	3.6
Community pharmacist	3	1.1
Pharmaceutical regulatory affairs	1	0.4
Other	19	6.9
Pharmacy years of experience; mean ± SD	8.0 ± 7.1	
Income in US dollar; mean ± SD	4364.5 ± 1829	
Weekly average working hours; mean ± SD	41.2 ± 10.2	

SD, standard deviation; US, United States.

**Table 2 healthcare-10-01441-t002:** Descriptive statistics and reliability of the used scales.

Items	Mean ± SD	Cronbach’s Alpha
Satisfaction	4.5 ± 1.3	0.89
Stress	2.7 ± 0.3	0.8
Expectation	3.9 ± 0.8	0.82
Performance	3.1 ± 0.9	0.82
Confirmation	−0.8 ± 0.27	0.82

**Table 3 healthcare-10-01441-t003:** Univariate linear regression analysis of predictors associated with pharmacists’ job satisfaction in Saudi Arabia.

Variable	Beta Coefficient	95% Confidence Interval		*p*-Value
		Lower	Upper	
Stress	−0.4408274	−0.5658094	−0.3158455	<0.001 *
Confirmation	0.3183	0.2077646	0.4288355	<0.001 *
Age	0.1641024	0.0515141	0.2766906	0.004 *
Gender				
Male	2.580723	1.163396	3.99805	<0.001 *
Female	Ref.			
Marital status				
Married	1.48	0.0093195	2.950681	0.049 *
Non-married	Ref.			
Income in US dollar Per 1000	0.3030119	−0.1548649	0.7608888	0.193
Having children				
Yes	1.340461	−0.0756364	2.756557	0.063
No	Ref.			
Chronic disease				
Yes	−1.522338	−3.358278	0.3136025	0.104
No	Ref.			
Education level				
Bachelor’s level degree	Ref.			
Graduate level degrees	1.788061	0.0573773	3.518745	0.043 *
Experience in years	0.1020107	−0.0003843	0.2044056	0.051
Total working hours/week	−0.0247226	−0.1009052	0.0514601	0.523
Hospital pharmacist				
Yes	−2.17619	−4.182874	−0.1695071	0.034 *
No	Ref.			

Note: *p*-value < 0.05 indicated with asterisk. Abbreviations: β, beta coefficient; 95% CI, 95% confidence interval; Ref., reference.

**Table 4 healthcare-10-01441-t004:** Multivariate linear regression analysis of factors associated with pharmacists’ job satisfaction in Saudi Arabia.

Variable	Beta Coefficient	95% Confidence Interval		*p*-Value
		Lower	Upper	
Stress	−0.4561981	−0.5616404	−0.3507558	<0.001 *
Confirmation	0.1479833	0.0059039	0.2900626	0.041 *
Age	0.1464493	−0.1290652	0.4219637	0.295
Gender				
Male	0.7298162	−0.9990614	2.458694	0.405
Female	Ref.			
Marital status				
Married	1.769961	−0.3662405	3.906162	0.104
Non-married	Ref.			
Income in US dollar Per 1000	−0.5548985	−1.173406	0.0636087	0.078
Having children				
Yes	−1.088697	−3.196374	1.018979	0.309
No	Ref.			
Chronic disease				
Yes	−0.7501685	−2.766525	1.266188	0.463
No,	Ref.			
Education level				
Bachelor’s level degree	Ref.			
Graduate level degrees	0.0531531	−2.216459	2.322765	0.963
Experience in years	0.0198419	−0.2505014	0.2901851	0.885
Hospital pharmacist				
Yes	−3.009126	−5.424853	−0.593398	0.015 *
No	Ref.			

Note: *p*-value < 0.05 indicated with asterisk. Abbreviations: β, beta coefficient; 95% CI, 95% confidence interval; Ref., reference.

## Data Availability

The data that support the findings of this study are available from the corresponding author.

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
