# Peer review of "Effect of Occupational Stress on Pharmacists’ Job Satisfaction in Saudi Arabia"

_healthcare, 2022, doi:10.3390/healthcare10081441_

Round 1

Reviewer 1 Report

Overall, this manuscript presents meaningful data and useful commentary on the pharmacist occupational stress and job satisfaction. I recommend that the manuscript be reconsidered and potentially accepted after major revisions to clarify study design and strengthen discussion arguments. Specific comments are numbered below:

Introduction

1. A nice introduction to the problem and need to assess work stress in pharmacists. The authors highlight the novelty of studying pharmacists in Saudi Arabia, however, I was missing justification for why studying this population (i.e.g, Saudi Arabian pharmacists) is important. The discussion contextualizes the results nicely within the environment, but I would have liked some of that information included in the background. For example: competition finding jobs, move to the private sector, policy impacting Saudi pharmacists.

Materials and Methods

2. Please add detail on how pharmacists were recruited to participate. Line 92 states data were collected from pharmacists in various sectors, how were they identified for the study? Emailed list of licensed pharmacists? Visiting places of employment? Social network?

3. Line 108 cites a previously published study which used the ERI instrument; please include this study citation. Currently citation 23 only links to instrument, not the study. 

Results

4. Line 173-174 states the "positive relationship" between pharmacists' job satisfaction and being married. With a p value = 0.049 (cutoff p < 0.05), I find this value only "slightly" significant and I'm not sure that it is a meaningly significant argument. I might exclude.

5. Given a majority of your sample worked in a hospital (80.6%), do you have a large enough sample to be able to state that job satisfaction was correlated with working in a hospital (e.g., do not have a large enough sample of pharmacists working outside of the hospital or in other settings)? I'd be concerned the large proportion of hospital pharmacists might bias results. This sentiment is also reflected in the discussion, below.

Discussion

6. Starting at line 244, I would need more information (i.e., study sufficiently powered to be able to detect change due to job) to feel comfortable with these discussion claims that "the present study found a negative association between pharmacists' job satisfaction and hospital satisfaction" (line 254). I think the following paragraph, starting at line 260, is interesting and still a worthwhile inclusion, but I might tie it in the general variability in hospital pharmacist satisfaction and the fact ~80% of the pharmacists in the study worked in a hospital setting, which is anticipated given the pharmacist workforce climate in Saudi Arabia. 

7. Minor wordsmithing; use of some subjective words throughout the discussion. Using "fair" as a descriptor of pharmacist work hours (line 208) and ECT as a "simple" concept (line 232). Would recommend removing phrases during editorial process. 

Reviewer 2 Report

Thank you for providing me with the opportunity to review the manuscript entitled “Effect of occupational stress on pharmacists’ job satisfaction in Saudi Arabia.” The following points should be addressed for further consideration in this journal.

Introduction

The authors should add information on the current role and work of pharmacists in Saudi Arabia in detail.

The study rationale should be presented appropriately.

Define confirmation.

Methods

The authors need to provide more information on the sample size calculation.

Mention the name of the research ethics committee with an approval number with the section of methods.

How data collectors were selected?

What was the qualification of the data collectors?

Was training provided to the data collectors?

How did data collectors’ approach potential respondents?

Who performed the back translation of the tool?

Which type of sampling technique was adopted?

What were the eligibility criteria for the selection of the respondents?

Have you checked the normality of the data?

Results

According to the information provided by the authors 'Saudi pharmacists in different sectors in Saudi Arabia, including academic institutions, hospitals, the drug industry, and conferences', this information should be presented in the demographics (Table 1).

Descriptive information about different tools utilized in this study should be presented as a supplementary file for a better understanding of the readers.

What have you done with the missing values?

Discussion

The discussion section should include examples of relevant studies undertaken in HICs.

The Limitations sections should be extended further.

Enlist a few recommendations of your study for policymakers.

Reviewer 3 Report

Overall, this is an interesting topic, relevant to the issue of job satisfaction, as well as burnout.

It will be useful to include the following:

- Descriptive summaries of survey question results (occupational stress, job satisfaction, expectation-confirmation)

- It might also be useful to reevaluate the covariates included in the multivariable analysis. I suspect that some were not confounding variables, and may not have needed to be included in the model. If you had a causal diagram, consider including as additional material.

- Throughout the document, and in the tables, consider changing "confirmation" to "expectation-confirmation" for clarity.

- In your multivariable analysis, given that stress is the primary variable, I am uncertain that interpreting expectation-confirmation and hospital employment should be interpreted in the same model, as all the covariates don't equally play in role in their relationships with job satisfaction.

Round 2

Reviewer 1 Report

Thank you for your revisions and addressing the comments!

Reviewer 2 Report

No further comments as authors have addressed all comments. Many congratulations to the authors.

Reviewer 3 Report

Thank you for making useful edits to the manuscript, and adding further clarifying information.